# Conformational Plasticity-Rigidity Axis of the Coagulation Factor VII Zymogen Elucidated by Atomistic Simulations of the N-Terminally Truncated Factor VIIa Protease Domain

**DOI:** 10.3390/biom11040549

**Published:** 2021-04-08

**Authors:** Jesper J. Madsen, Ole H. Olsen

**Affiliations:** 1Global and Planetary Health, College of Public Health, University of South Florida, Tampa, FL 33612, USA; 2Novo Nordisk Foundation Center for Basic Metabolic Research, Section for Metabolic Receptology, University of Copenhagen, Blegdamsvej 3b, DK-2200 Copenhagen, Denmark

**Keywords:** zymogen, structure-function relationship, allostery, tissue factor, molecular dynamics, replica-exchange

## Abstract

The vast majority of coagulation factor VII (FVII), a trypsin-like protease, circulates as the inactive zymogen. Activated FVII (FVIIa) is formed upon proteolytic activation of FVII, where it remains in a zymogen-like state and it is fully activated only when bound to tissue factor (TF). The catalytic domains of trypsin-like proteases adopt strikingly similar structures in their fully active forms. However, the dynamics and structures of the available corresponding zymogens reveal remarkable conformational plasticity of the protease domain prior to activation in many cases. Exactly how ligands and cofactors modulate the conformational dynamics and function of these proteases is not entirely understood. Here, we employ atomistic simulations of FVIIa (and variants hereof, including a TF-independent variant and N-terminally truncated variants) to provide fundamental insights with atomistic resolution into the plasticity-rigidity interplay of the protease domain conformations that appears to govern the functional response to proteolytic and allosteric activation. We argue that these findings are relevant to the FVII zymogen, whose structure has remained elusive despite substantial efforts. Our results shed light on the nature of FVII and demonstrate how conformational dynamics has played a crucial role in the evolutionary adaptation of regulatory mechanisms that were not present in the ancestral trypsin. Exploiting this knowledge could lead to engineering of protease variants for use as next-generation hemostatic therapeutics.

## 1. Introduction

Coagulation factor VIIa (FVIIa) is a protease that initiates the enzymatic cascade of reactions leading to a burst of thrombin activation and blood clot formation [1,2,3,4]. FVIIa consists of an N-terminal membrane-binding domain rich in γ-carboxyglutamic acid residues (Gla domain), two epidermal growth factor-like domains (EGF1 and EGF2), and a trypsin-like serine protease (SP) domain, which is responsible for performing the enzymatic function [3]. Like trypsin and most other trypsin-like proteases, FVIIa is produced as a single-chain zymogen, FVII, that requires proteolytic cleavage for activation. However, unlike trypsin, FVIIa is a catalytically incompetent protease on its own, relying on the cofactor, Tissue Factor (TF), as an allosteric activator. While numerous distinct structural aspects of FVIIa affect its biochemical and physiological functions [5,6,7,8,9,10,11,12,13,14,15,16], the focal point is the SP domain when it comes to the activation transition. The activation process is thoroughly studied and largely understood in the trypsinogen-to-trypsin transition [17,18,19,20,21]: proteolytic cleavage before position I16 (chymotrypsin numbering) produces a new N-terminal tail that spontaneously inserts into the activation pocket (or “Ile cleft”) and forms a salt bridge with D194, leading to a maturation of the S1 pocket and catalytic competency. In addition, three loop segments, activation loop 1, 2, and 3 (AL1–3), undergo stabilization [19,22]. The N-terminal insertion of trypsinogen activation has even been observed previously using accelerated molecular dynamics simulations [23]. While the activation transition of FVII is known to follow the same hallmark steps as trypsinogen-trypsin for it become catalytically active, many important details concerning FVII and the zymogen-like FVIIa are lacking.

The structural and dynamic details of FVII before, during, and after each of the two activation transitions (proteolytic and allosteric) are of considerable interest. Structures of FVIIa [24] and the FVIIa-TF complex [25,26] have been tremendously helpful in elucidating the molecular basis of hemostasis. Unfortunately, key forms of this important protease (including FVII and the free, uninhibited FVIIa in the zymogen-like form) have remained elusive despite heroic efforts by expert crystallographers over the last several decades. Modeling and simulation efforts have been employed to compensate for the lack of these structures with some success [27,28].

The native trypsin does not undergo conformational or dynamic changes upon binding of active-site inhibitors [29]. However, other trypsin-like proteases that are regulated by protein cofactors, e.g., FVIIa or thrombin [30], rely critically on a dynamic apo state, presumably because the inherent conformational flexibility allows for a mechanism of allosteric activation by the cofactor(s). Extensive experimental and computational investigations [28,31,32,33] have resulted in the hypothesis that two primary allosteric pathways mediate TF-induced enhancement on FVIIa activity [34,35]. One pathway tethers the 170-loop to the SP domain by stabilizing the a1 helix (a.k.a. the TF-binding helix) and enables G223 to form a hydrogen bond with the mainchain nitrogen of R170C [28,36], which restrains the flexibility of W215 in the substrate pocket S2-S3. The other pathway, on the other hand, travels from L163 to F225, stabilizing AL3 and, via. S185, AL2. This facilitates the insertion of the N-terminus and the proper salt bridge formation between I16 and D194. Knowledge about the activation mechanism and allostery has already yielded successful engineering of FVIIa variants with enhanced activity independent of stimulation by endogenous TF [37,38,39]. Of particular interest is a recently engineered FVIIa-trypsin variant, FVIIa_VYT_, which is completely TF-independent due to its grafted trypsin 170-loop (as previously described [40,41,42]) in addition to another mutation, L163V [39]. The grafted trypsin 170-loop and the single mutation seem to stabilize the first pathway and second pathway, respectively.

In this work, we employ a computational modeling and simulation approach to generate a faithful realization of FVII by N-terminal truncation of FVIIa. The method is validated by demonstrating that it can be used meaningfully for trypsin and that the N-terminally truncated variant, trypsin-desIVG, adopts the structural and dynamic characteristics of trypsinogen. To our knowledge, this is the first report of successful inactivation of a trypsin-like protease by truncation of the inserted N-terminal tail.

## 2. Materials and Methods

### 2.1. Molecular Dynamics (MD) Simulations

#### 2.1.1. Temperature-Replica-Exchange Molecular Dynamics (T-REMD) Simulations

T-REMD [43] simulations of the trypsin-like SP domains were performed using Gromacs 5.0.4 [44] with the CHARMM36m force field [45]. Ca^2+^ bound to the calcium-binding loop was kept while crystal waters, inhibitors, and other co-crystallized moieties were removed in the protein preparation step. Explicit TIP3P water [46] with 150 mM KCl was used for solvation. A 12-Å cut-off for the van der Waals forces were used. Electrostatic forces were computed using the particle mesh Ewald method [47]. The Verlet cut-off scheme was used. A total of 10 replicas were swapped in a temperature range of 310.15 K (physiological temperature) to 331.21 K (lower than the protein denaturing temperature) in steps of *∆T =* 2.34 K. Systems were initialized with random velocities at the reference temperatures. The temperature and pressure were controlled using the Nosé-Hoover [48,49,50] and Parrinello-Rahman [51,52] methods, respectively, to sample the *NPT* ensemble at 1 bar and the reference temperature for the replica. The integration time step was 2 fs, enabled by using H-bond restraints [53]. Exchange of adjacent replicas was attempted every 500 steps (i.e., every 1 ps) and the average exchange acceptance probability was ~0.20 for all systems. The T-REMD simulations were carried out for 100–200 ns per replica (200 ns for FVIIa and FVIIa-desIVG; 100 ns for the rest), resulting in a combined simulation time of 1–2 μs/system. Only the physiological temperature (310.15 K) replicas were used for analysis of structural and dynamic properties of the proteins. An overview of the systems simulated can be seen in Table 1. These systems were chosen for the following reasons: trypsin(ogen) would allow benchmarking our approach using a system that was extensively studied and where the zymogen structure was known. This gave confidence to applying the computational method to the study of FVII(a) (and a key variant hereof), where the zymogen remained elusive despite significant experimental efforts. The FVIIa_VYT_ variant was chosen in addition because it was a soluble tissue factor-independent variant of FVIIa that was functionally the closest to that of trypsin. On the spectrum from FVIIa to trypsin, the FVIIa_VYT_ variant was therefore presumed to be the most trypsin-like construct among engineered variants of FVIIa in terms of dynamics and functional characteristics.

#### 2.1.2. The In Silico Zymogenized Variants

The trypsin-like proteases were zymogenized in silico to give constructs without the inserted N-terminal tail, thus mimicking the zymogen (denoted by the term “desIVG”). This was done by truncation of the SP domain N-terminal by removing the leading three amino acid residues (^16^IVG^18^). The new N-terminal residue, G19, was neutralized by amidation to avoid local artifacts caused by the presence of the positive charge. The truncation approach was found preferable when compared with applying a force (e.g., using steered MD) to pull the N-terminal tail out of the activation pocket because of the salt bridge between I16 and D194, which tended to persevere and cause the structure to unravel as the I16-D194 pair was being pulled.

### 2.2. Analyses

#### 2.2.1. Root-Mean-Squared Deviation (RMSD) and Root-Mean-Squared Fluctuation (RMSF)

RMSD calculations were done with PyMOL’s align command without outlier rejection (cycles = 0). RMSF calculations were done using gmx rmsf (taking contributions from each atom in the residue selection) and the simulated B-factors (Debye-Waller factors) were calculated as [56]:(1)$$B_{i}=\frac{8\pi^{2}}{3}{\langle MSF_{i}\rangle }_{\mathrm{MD}}$$

#### 2.2.2. E/Z-E*/Z* Distributions

E/Z vs. E*/Z* are distinguished by a specific structural segment (the 215–217 segment) in the SP of the trypsin-like proteases: One that is fully accessible to the substrate (E/Z) and the other occluded by collapse of the 215–217 segment. The conformations were categorized with a distance-based cut-off criterion. For E/Z, the defining distances are W215/CG to H57/CG > 6.5 Å and W215/CG to S195/OG > 6 Å; otherwise the configuration is designated E*/Z*.

#### 2.2.3. Principal Components Analysis (PCA)

PCA is a useful technique for reducing the dimensionality of MD simulation data and extracting larger amplitude motions. The method works by computing the eigenvectors of the mass-weighted covariance matrix of the atomic positional fluctuations (here we used all heavy atoms) [57]. To perform the PCA analysis, we used pytraj [58] and scikit-learn [59].

#### 2.2.4. Hierarchical Clustering Analysis

Structural clustering analysis allowed us to assign different structures observed during the MD simulations into distinct groups. The distance metric used was the pairwise RMSD and we performed the clustering on the *C_α_* subset of atoms. Agglomeration was performed using the average/UPGMA linkage approach [60]. Pytraj [58] and SciPy [61]/NumPy [62] were used to perform the clustering analysis.

#### 2.2.5. Dynamic Cross-Correlations

Dynamic cross-correlations between the atoms within the molecule is a measure of the extent to which they move together in a correlated manner. The values in the resulting NxN heatmap (N = the number of *C_α_* atoms) correspond to the dynamic cross-correlation between each (i,j) atom and a correlation of 1(–1) indicates complete (anti)correlation. A value of 0 indicates no correlation exists. To perform the dynamic cross-correlation anaylsis, we used MD-TASK [63]/MDTraj [64].

#### 2.2.6. Plotting and Visualization

Inspection of structures and trajectories was done using VMD [65] and PyMOL (Schrödinger LLC, Open-Source version 2.3.0). Plotting was done using matplotlib [66] and Xmgrace (Grace version 5.1.25). Figure layout was done with Inkscape (http://www.inkscape.org/, 1 February 2021, version 1.0).

## 3. Results and Discussion

### 3.1. The N-Terminally Truncated Trypsin Gradually Adopts Structural and Dynamic Features Characteristic to Trypsinogen

The results and conclusions presented in this work rely on the application of an in silico protocol for interrogating the conformational structure and dynamics of the inactive zymogen precursor of a trypsin-like protease. Specifically, the protocol involves the truncating the protease domain N-terminal by removing the leading three amino acid residues (^16^IVG^18^) and subjecting the inactivated construct to MD simulations. Here, we adopted a replica-exchange scheme to enhance sampling of the MD simulation, a popular approach in the study of biomolecular systems where an appropriate collective variable (CV, or reaction coordinate) is not readily defined. Defining a CV for biasing a trypsin-like protease toward its zymogen structure presupposes deep knowledge about the system, locally and globally. Intuitively, one could in principle expel the N-terminal tail from the activation pocket by applying a local bias e.g., along a vector projection on the exit path. However, this would not necessarily allow the resulting structure and dynamics of the protein to approach the zymogen-like state faster (in a global sense) than by performing regular MD simulations of the N-terminally truncated construct, as non-local slow degrees of freedom would still hamper the transition. To best handle these challenges, we choose to combine N-terminally truncation with an enhanced sampling technique that facilitates global configurational exploration, namely T-REMD.

To validate our protocol, we first applied it to the evolutionary ancestor of FVIIa, trypsin, and the trypsinogen-trypsin activation transition. A straightforward way to assess whether the structural characteristics of the simulated trypsin-desIVG leans more toward trypsin or trypsinogen is to compute the root-mean-square deviation (RMSD). We observed that while trypsin-desIVG initially retained its trypsin-like characteristics on a global RMSD-based scale, it quickly (in ~20 ns) transitioned into, and remained in, a conformation more closely resembling the simulated average structure of trypsinogen (Figure 1), albeit with significant structural fluctuations. This was remarkable because it happened despite trypsin-desIVG being initialized from the actual trypsin structure (except the truncated N-terminus) and was compared with the simulated average structures of its structural reference and another crystal structure entirely (namely trypsinogen). However, recall that the MD-averaged structures are generated by positional averaging and does therefore not correspond to a structural realization of the conformational ensemble. This is because positional averaging will not, in general, preserve all bond lengths, angles, etc. A matrix for computed RMSD values among all simulated average structures and reference crystal structures used in this study is available in Appendix A (Appendix A).

An analysis of the per-residue root-mean-square fluctuation (RMSF) of the three simulated trypsin(ogen) constructs reveals that trypsin is very rigid with only minor flexible segments distributed throughout (e.g., at the tip of certain loops). Trypsin-desIVG and trypsinogen, on the other hand, exhibit a practically identical extended spatial region of high flexibility (Figure 2a,b). This flexible region encompasses activation loops 1 (residues 144–152), 2 (residues 184–193), and 3 (residues 221A–226) (AL1–3). Notice that the flexibility of the 170-loop is comparably low for all three trypsin(ogen) constructs. It is useful to further compare this with the experimentally determined B-factors from crystallography, which follow the same trends, albeit with much lower flexibility altogether due to the cryogenic temperatures and lattice conditions subjected to a protein in a crystal (Figure 2c).

### 3.2. Conformational Flexibility of FVIIa, the TF-Independent Variant FVIIa_VYT_, and Trypsin

The term flexibility refers here to the fast and reversible changes the system undergoes. The per-residue RMSF can be used as a measure to gauge the local conformational flexibility. A comparison between the three SP domains and their N-terminally truncated counterparts reveals that the activation pocket collapses (Appendix A), and that the surrounding segments (AL1–3 and S1 pocket) dramatically increase their structural flexibility as a result, when the inserted N-terminal tail is removed (Figure 2a,b). Such an increase is observed in all zymogen-enzyme pairs. It is thus expected by be shared by most trypsin-like zymogens based on the current results as well as available crystallographic evidence [67]. In addition, we observed that the flexibility of the 170-loop was unaffected by the N-terminal insertion and the resulting structural maturation of the AL1–3 and S1 pocket, suggesting that a shortened 170-loop conferred intrinsic stability of the trypsin-like proteases but not vice-versa. It is informative to compare the flexibility of each structural element (e.g., 170-loop or AL1–3; see also Figure 1a and Appendix A) between the enzyme construct (FVIIa, FVIIa_VYT_, trypsin) and the corresponding N-terminally truncated construct (-desIVG). The characteristics of, and differences between, the trypsinogen-trypsin and FVII-FVIIa conformational transitions are striking. Even though the percent sequence identity between trypsin/FVIIa and trypsin/FVIIa_VYT_, are 41% and 42%, respectively [68], the carefully designed mutations of FVIIa_VYT_ (i.e., by grafting the trypsin 170-loop and L163V) manage to shift the conformational characteristics of the FVIIa_VYT_ SP domain to effectively mimic trypsin as gauged by local flexibility (Figure 2a,b). In particular, notice how the flexible regions (Figure 2b, red areas) throughout the FVIIa_VYT_ molecule matches up with the pattern observed in trypsin, not with that of FVIIa. This is clearly the case for the 170-loop, but closer inspection reveals that it also applies for the 215–217 segment that harbors W215. As we shall see in the following sections, various other structural and dynamic analyses of the conformational ensembles are consistent with this hypothesis.

In general, there is a similar distribution of flexible regions in the zymogen-like states and the active site triad conformations are preserved during the step of proteolytic activation, with FVIIa and FVIIa-desIVG exhibiting more variation in either measure. N-terminal tail insertion induces (directly or indirectly) the catalytically competent conformation by maturing the S1 pocket, AL1–3, and the oxyanion hole of the zymogens. This process is evidently accompanied by substantial conformational changes and changes in flexibility as well. The disorder-to-order transition is likely an essential regulatory mechanism for preventing premature proteolysis in more advanced organisms. The zymogen-like property of FVIIa may have its origins in a self-inhibitory property due, at least in part, to the outstanding conformational flexibility. It is unclear whether this regulatory framework is evolutionarily conserved among all species carrying FVIIa, something that has recently sparked some debate [69,70,71]. From the literature, we know that the energetic origins of the conformational changes induced by N-terminus insertion are both electrostatic and hydrophobic [23,72,73]. Thus, the zymogen-to-enzyme activation transition can be considered a cooperative process with distinct energetic and entropic contributions, making it difficult to delineate adequately on the basis of static structures. It is worth mentioning, however, that we were recently able to rationalize important aspects of FVIIa allostery by using the ensemble-refinement technique [74,75] that combines X-ray structure refinement with molecular dynamics in order to build an ensemble of structures consistent with the electron density map [76].

### 3.3. The Allosteric E-E* and Z-Z* Equilibria Are Influenced by the 170-Loop, but Independent of N-Terminus Insertion

Structural and biochemical investigations of thrombin, another trypsin-like protease, has suggested that a structural segment around residues 215–217 exhibits distinct conformations of direct functional relevance [30,77,78,79,80,81,82]. The terminology used here, E(Z) and E*(Z*), refer to the enzyme (E) or zymogen (Z) 215–217 segment being in the trypsin-like matured conformation (E/Z) or the substrate-occluding conformation (E*/Z*). Recent evidence supports that a similar conformational change is relevant for FVIIa [28,33,42]. In particular, the 215–217 segment has been implicated in substrate accessibility to the active site and cofactor-mediated allosteric regulation [28,42]. We have previously used MD simulations to successfully explore the conformational equilibrium of the 215–217 segment in variants of FVIIa [42].

Surprisingly, the current work suggests that proteolytic (in)activation and N-terminus insertion has little to no influence on E-E* and Z-Z* equilibria in any of the investigated protease variants (Figure 3a,b). FVIIa and FVIIa-desIVG have similarly substrate-occluding 215–217 segment conformations (Figure 3a). We have previously shown that FVIIa shifts its population from E* to E when complexed with TF [42] and we speculate that the same might be true for FVII. The time evolution of the E/Z-E*/Z* states of the individual constructs show that transitions are intermittent and plentiful for FVIIa and FVIIa-desIVG, which are the two constructs that regularly visits both their respective E/Z and E*/Z* conformations (Appendix A). We find that the characteristic behavior of FVIIa_VYT_(-desIVG) is effectively shifted toward that of trypsin(ogen), which is essentially locked in the E(Z) configuration (Figure 3b,c). These results suggest that the allosteric equilibria of the 215–217 segment is governed by the 170-loop, independently of N-terminus insertion, supporting the notion that W215 in the 215–217 segment is the endpoint of the proposed first allosteric pathway [35] alluded to the introduction.

### 3.4. Conformational Plasticity-Rigidity Axis Intrinsically Governed by the 170-Loop with the N-Terminus Insertion as Extrinsic Modulator

There are distinct differences in both structure and dynamics of FVIIa, FVIIa_VYT_, and trypsin. In the above (Section 3.2), we address the conformational flexibility of the simulated proteins and variants by assessing structural fluctuations. Here, we look into notions of plasticity and rigidity, which are increasingly appreciated as essential for elucidating structural and functional characteristics of proteins [21,83,84,85,86,87,88]. Though, the terms are ill-defined despite being commonly used. We adopt the definition by Csermely, who propose that plasticity and rigidity are opposing characteristics of the system due to external or internal changes [89]. Structural plasticity and rigidity are defined in terms of structural properties and they need not necessarily be synonymous with their functional counterparts.

We performed PCA of the simulations of the three proteases and their corresponding zymogen counterparts to complement the structural analyses (Figure 4). PCA is useful technique for reducing the dimensionality of MD simulation data and extract larger amplitude motions and, as such, allows us to concretely compare the dynamics exhibited by the proteins in the essential subspace [57,90]. The analysis reveals that the simulated conformational ensembles for FVIIa and FVIIa-desIVG exhibit much larger subspace variation; even more so than that of trypsinogen, which is the least compact of the rest. Here, we use the term “compactness” to visually assess the PCA analyses. Used this way, (lack of) “compactness” is consistent with conformational rigidity (plasticity) as defined and used in the current work. Generally, we find that the constructs can be ordered in terms of their subspace compactness as follows (from most to least): trypsin, FVIIa_VYT_, trypsin-desIVG, FVIIa_VYT_-desIVG, trypsinogen, FVIIa, and FVIIa-desIVG (Figure 4). The unambiguous trends that emerge by inspection of the distributions of the first two principal components are that (1) truncating the N-terminal tail increases subspace variability (with trypsin-desIVG approximating trypsinogen), (2) FVIIa_VYT_ effectively adopts the dynamical characteristics of trypsin, and (3) FVIIa/FVIIa-desIVG exhibits the most subspace variability by far. Furthermore, the gaussian and unimodal nature of the PC1-PC2 scatter plots for FVIIa indicates the presence of substantial conformational plasticity of FVII/FVIIa. The compact, non-gaussian and multimodal nature of FVIIa_VYT_ and trypsin, on the other hand, indicates conformational rigidity. Recall that major conformational changes between distinct conformational states are expected to be captured reasonably well by the first few PCs while the dynamics of a rigid protein in a narrow energetic well may involve independent motions of many different parts of the protein and, as such, exhibit higher dimensionality in the essential dynamics space as well as local frustration [57,90,91,92].

These observations demonstrate that the in silico zymogenization method we apply enables us to transform the SP domains into their corresponding zymogens, not just measured by calculating the RMSD to a reference structure (as shown in Figure 1), but also in the essential dynamics subspace (Figure 4). Taken together with considerations of molecular flexibility (Figure 2), these analyses are effective in elucidating the structural and dynamic features of the conformational ensembles. In addition, there is some evidence from the current work that inactivation by N-terminus truncation gives rise to a bifurcated conformational ensemble for trypsin and FVIIa_VYT_, but for not for FVIIa. This finding comes about by performing RMSD-based hierarchical clustering on the simulated ensemble of structures and characterizing the number of clusters using a cut-off or a dendrogram (Appendix A). The increase in the number of distinct clusters after N-terminal truncation for trypsin and FVIIa_VYT_, and lack hereof for FVIIa, is consistent with the rigidity of trypsin and FVIIa_VYT_, and the plasticity of FVIIa. However, the choice of clustering method has some influence on the results. To further support the idea of different occurrence of structural clusters in the simulated ensembles between trypsin/FVIIa_VYT_ and FVIIa, we computed per-residue dynamic cross-correlations over the individual trajectories. The dynamic cross-correlations were curiously similar for all cases, suggesting that the dynamic footprints of a bifurcated conformational ensemble may be averaged out over the sampled time scale (Appendix A). It is conceivable that the similarities among the cross-correlation heatmaps are at least in part due to the frequent and constant exchanges between replicas.

It’s remarkable that the engineered variant, FVIIa_VYT_, clearly exhibits the characteristics of trypsin in excellent agreement with experimental measurements, reassuring us that this variant effectively “beats TF at its own game” as proclaimed [39]. Concretely, this means that FVIIa_VYT_ exhibited 60-fold higher amidolytic activity than FVIIa, and displayed similar factor X activation and antithrombin inhibition kinetics to the FVIIa-sTF complex [39]. It is perplexing, however, that the chemical reactivity of the N-terminal tail of FVIIa_VYT_ (gauged via the carbamylation assay [93]) falls strictly in-between that of FVIIa and FVIIa-sTF, with a three-fold higher KNCO t_1/2_ than the former and a four-fold lower KNCO t_1/2_ than the latter. Our results do not resolve this apparent paradox and further investigations are needed. In particular, it remains unclear from a structural point of view exactly how KNCO chemically reacts with the N-terminal primary amine of I16. The trypsinogen-to-trypsin transition would suggest that the N-terminal tail is available for chemical modification only once expelled from the activation pocket and extruding outward from the SP domain body (in a trypsinogen-like configuration [55]). However, it might be possible for water and KCNO to access the N-terminal residue between the loop segment that forms the rim of the S1 pocket (residues 190–195) and AL1. MD simulations have previously indicated that solvation and destabilization of the I16-D194 salt bridge can occur by this mechanism [28] and it would be interesting to look further into this aspect using specialized analysis methods [94,95,96] or collective-variable (CV)-based enhanced sampling methods such as metadynamics [97]. This would be particularly pertinent in relation to assessing the solvent accessibility of I16 in different conformational states of FVIIa.

### 3.5. A Glimpse of the Elusive FVII Zymogen Exposed at Last?

The structural basis for the functional and biochemical similarities between the zymogen-like FVIIa and FVII is poorly understood. The main reasons are that key forms of this important protease (including FVII and the free, uninhibited FVIIa in the zymogen-like form) have remained elusive despite heroic efforts by expert crystallographers. An eccentric activation mechanism of FVII involving reconfiguration of β-strands of the protein structural scaffold has been proposed based on a crystal structure [9]. However, detailed experimental and theoretical investigations [31,98,99] do not support the reregistration hypothesis. Remarkably, a structure of the free FVIIa without an inhibitor has been obtained by soaking out the inhibitor from benzamidine-inhibited FVIIa crystals [24]. While this structure alludes to added flexibility of FVIIa in the absence of inhibitor, it does not resolve the issues relating to the zymogen-like character of the free and uninhibited FVIIa in solution.

It is a reasonable assumption that the functional similarities between the zymogen-like FVIIa and FVII arise due to their structural and dynamic similarities. This work sheds some light on the properties of both FVIIa and FVII insofar that FVIIa-desIVG is a reasonable surrogate for FVII, which we have argued in the above. It must be emphasized, however, that it is not known how many distinct conformational basins (or dominant conformational intermediates) there are in the activation process from FVII via FVIIa to FVIIa-TF. Our results demonstrate the vivid flexibility of parts of the FVIIa-desIVG molecule (Figure 2 and Figure 3) and the inherent plastic character of its conformational ensemble (Figure 4). This is consistent with the view that FVII may not have a dominant, well-defined structure in large parts of the molecule.

It is worth noting that even in cases where the zymogen structures of advanced trypsin-like proteases are available (such as profactor D [100] and factor XI [101]; see the review by Gohara and Di Cera for an overview of available zymogen structures of trypsin-like protases [67]), a recurring theme is that the parts of the structures that we are most interested in are too flexible and divulge insufficient electron density for reliable model building.

## 4. Conclusions

We report MD simulations of FVIIa, FVIIa_VYT_ (a TF-independent variant), trypsin(ogen) and their N-terminally truncated counterparts. By employing the replica exchange method for improved sampling, we have explored the conformational dynamics of these protease constructs at effective time scales beyond that what the corresponding conventional MD simulations would have explored.

Our results demonstrate the feasibility of a in silico zymogenization protocol that involves N-terminal truncation of the three leading amino acid residues of the activated protease to produce a construct that takes on the structural and dynamic characteristics of the zymogen. The method is used to shed light on the structure and dynamics of the elusive FVII zymogen. By comparing and contrasting with trypsin and the TF-independent FVIIa variant, FVIIa_VYT_, we show that the conformational plasticity-rigidity axis is primarily governed by the 170-loop and modulated by N-terminal tail insertion. Conformational flexibility is affected mostly by N-terminal tail insertion. Finally, the allosteric E-E* and Z-Z* equilibria are influenced by the 170-loop, but independent of N-terminal tail insertion.

That a relatively minor modification in the 170-loop can make FVIIa TF-independent likely means that TF’s role in supporting the zymogen-like FVIIa is compensating for an effect that the trypsin 170-loop supplies to the SP domain in terms of flexibility and rigidity (but that the FVIIa 170-loop fails to). Considering that N-terminal tail insertion is a mechanistic step conserved among other trypsin-like proteases, this naturally restricts the space of evolutionary possibility to making adaptations that utilize the conformational plasticity-rigidity axis.

## Figures and Tables

**Figure 1 biomolecules-11-00549-f001:**
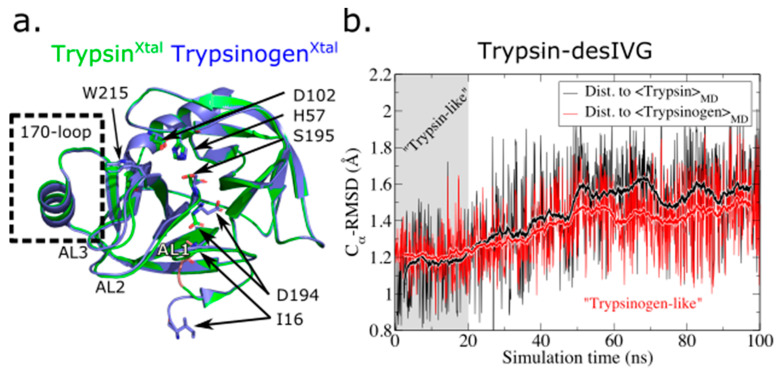
(**a**) Structural comparison of the trypsin and trypsinogen crystal structures. Select amino acid residues and spatial regions of special interest are labeled. Structure PDB IDs are indicated in Table 1. (**b**) The alpha-Carbon root-mean-squared deviation (*C_α_*-Root-mean-squared deviation (RMSD)) from the simulated average structures of trypsin (<Trypsin>*_MD_*) and trypsinogen (<Trypsinogen>*_MD_*) is plotted for the trypsin-desIVG trajectory. The thicker lines with white edges indicate the running-average value over 50 steps.

**Figure 2 biomolecules-11-00549-f002:**
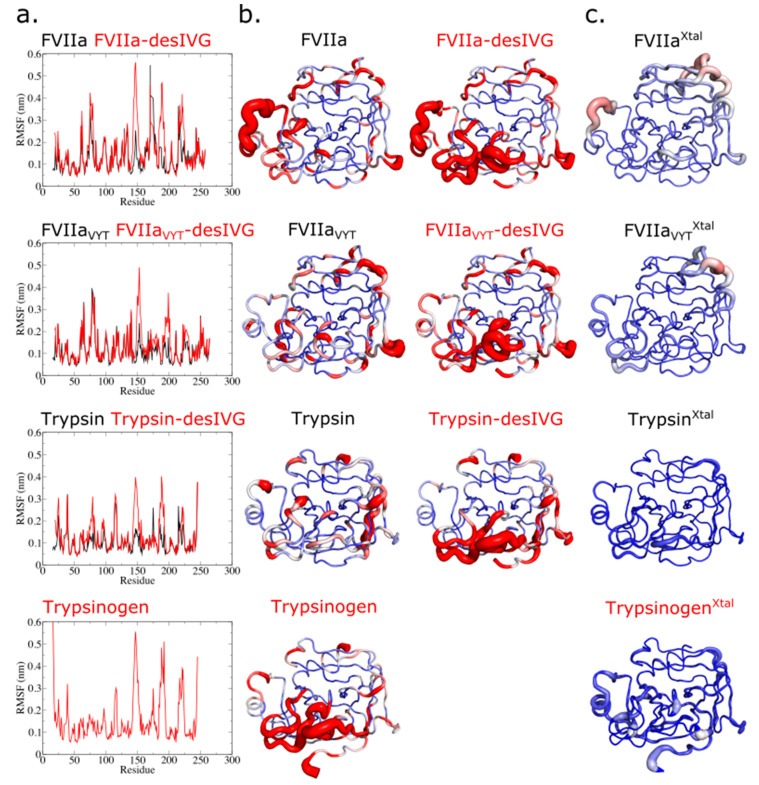
(**a**) Per-residue root-mean-square fluctuation (RMSF) of the simulated constructs. (**b**) Visualization of the B factors based on the computed RMSF values. Proteins are shown using a “sausage” cartoon representation where the thickness of the tube indicates the B-factor. The color scale goes from low (blue) over medium (white) to high (red). (**c**) Visualization of the crystallographic B-factors. Proteins are shown using a “sausage” cartoon representation where the thickness of the tube indicates the B-factor. The color scale goes from blue (low) over white (medium) to red (high). Structure PDB IDs are indicated in Table 1.

**Figure 3 biomolecules-11-00549-f003:**
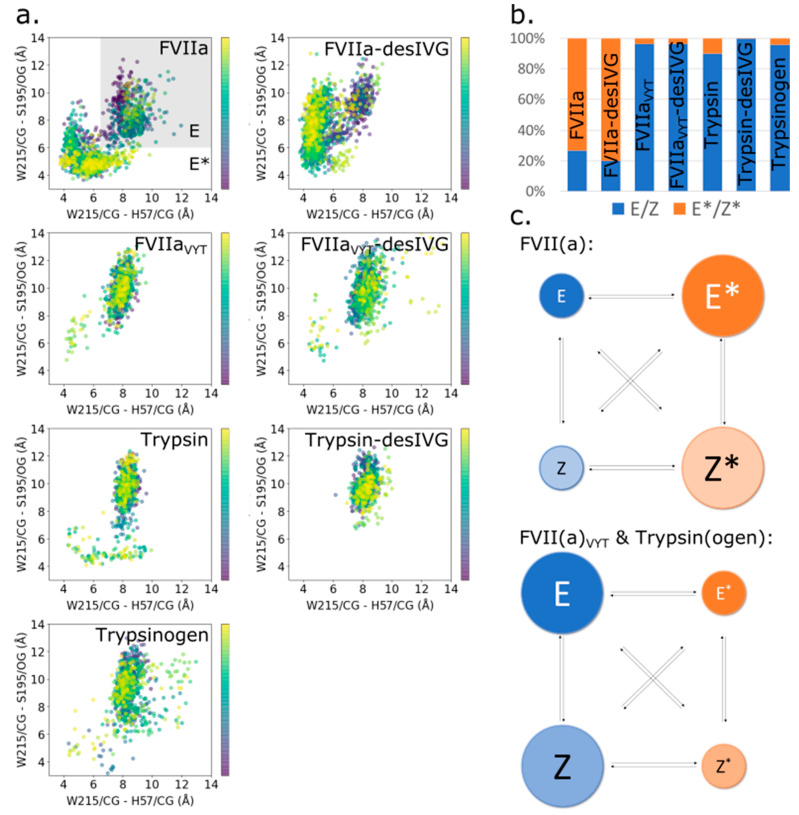
(**a**) Scatter plots of the W215 location relative to the active site residues H57 and S195 of the simulated constructs. The color bar scale indicates the simulation time starting from 0 ns (purple dots) to the end of the trajectory (yellow dots). (**b**) Quantification of E-E*/Z-Z* equilibria from (**a**) based on the cut-off criterion. Errors in the estimates based on bootstrapping with 1000 samples were <1% and errorbars showing the bootstrap standard error are therefore too tiny to be seen. (**c**) Schematic depiction of relative populations of E-E*/Z-Z*. FVII(a) is found predominantly in Z*/E* configurations, while FVII(a)_VYT_ and trypsin(ogen) are found to almost entirely occupy Z/E configurations. Size ratios between the spheres are indicative only and not to scale; see the histogram in (**b**) for accurate quantification.

**Figure 4 biomolecules-11-00549-f004:**
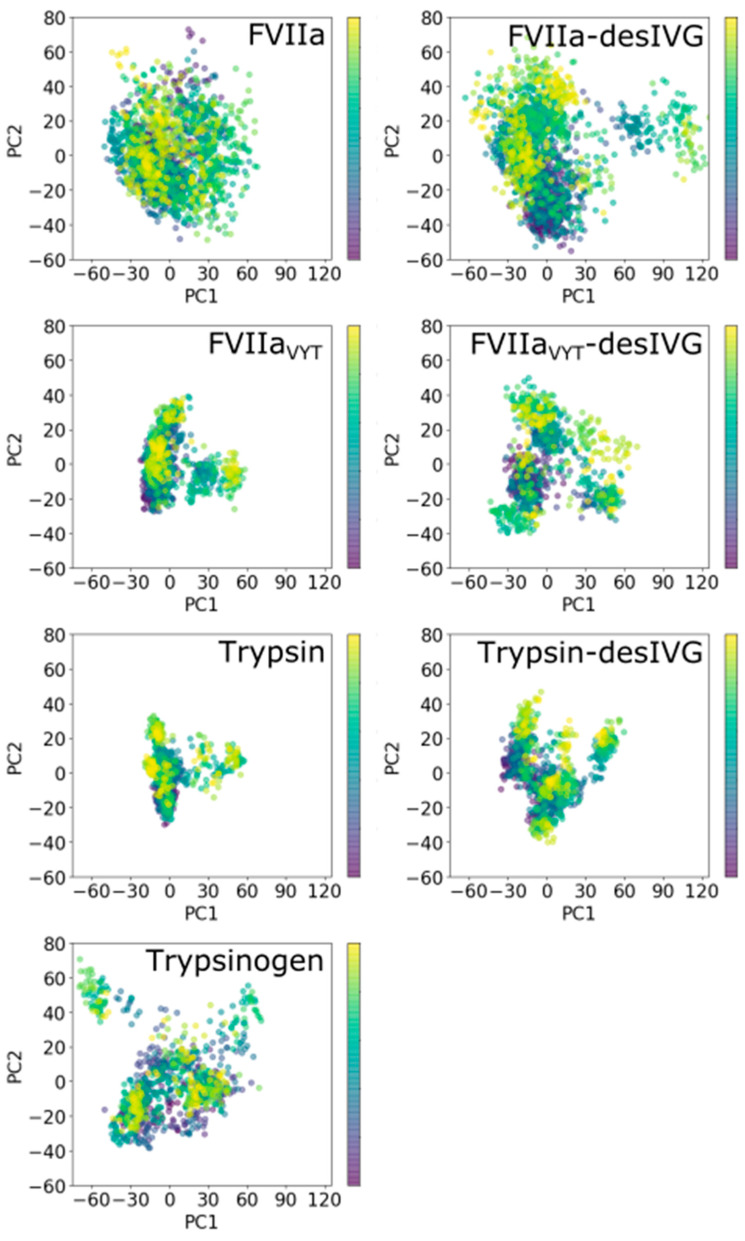
Principal component analysis of the simulated constructs. The plots show the first (PC1) and second (PC2) principal component against each other. The color bar scale indicates the simulation time starting from 0 ns (purple dots) to the end of the trajectory (yellow dots).

**Table 1 biomolecules-11-00549-t001:** Summary of the structures used, and the systems simulated in this study.

Name	Description	PDB ID	Simulation Protocol	Simulation Time (μs)
FVIIa	Human FVIIa SP domain.	1dan_H [25]	T-REMD	2
FVIIa-desIVG	Human FVIIa SP domain. In silico zymogenized.	1dan_H [25]	T-REMD	2
FVIIa_VYT_	Human FVIIa_VYT_ SP domain, a TF-independent variant of FVIIa.	6r2w_H [39]	T-REMD	1
FVIIa_VYT_-desIVG	Human FVIIa_VYT_ SP domain, a TF-independent variant of FVIIa. In silico zymogenized.	6r2w_H [39]	T-REMD	1
Trypsin	Bovine trypsin.	1j8a [54]	T-REMD	1
Trypsin-desIVG	Bovine trypsin. In silico zymogenized.	1j8a [54]	T-REMD	1
Trypsinogen	Bovine trypsinogen.	2tgt [55]	T-REMD	1

FVIIa, coagulation factor VIIa; desIVG, N-terminally truncated construct; SP, serine protease; TF, tissue factor; T-REMD, Temperature-replica-exchange molecular dynamics.

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
