# Peer review of "Conformational Plasticity-Rigidity Axis of the Coagulation Factor VII Zymogen Elucidated by Atomistic Simulations of the N-Terminally Truncated Factor VIIa Protease Domain"

_biomolecules, 2021, doi:10.3390/biom11040549_

Round 1
Reviewer 1 Report
In the manuscript titled “Conformational Plasticity-Rigidity Axis of the Coagulation Factor VII Zymogen Elucidated by Atomistic Simulations of the N-terminally Truncated Factor VIIa Protease Domain”, the authors performed a series of molecular dynamics simulations to characterize the coagulation factor VII, in its zymogenic and active forms. They explore the conformational plasticity of the protease domain of the zymogenic FVII with a comparative set of calculations on another coagulation protein trypsin and its zymogenic form trypsinogen that was well characterized experimentally. They tested their hypothesis on an “in silico” zymogenized variant created by removing 3 N-terminus residues of the protease domain of trypsin and comparing the dynamical characterization of that system with the dynamics of the true zymogenic form for which the X-ray crystal structure was available. After establishing this truncation approach in constructing a zymogenic mimic, the authors applied the methodology in creating a complementary construct for zymogenic FVII using the structure of FVIIa and followed its dynamics. They also, carried out simulations of the TF-independent variant of FVIIa and its “in silico” zynogenized form. The methods employed in performing simulations and the subsequent analysis of the trajectories were well executed and the manuscript was well written.
From the results of various analysis formed on above systems, the authors established the feasibility of their "in silico" zymogenization protocol. With the comparison of the results from trypsin (both active and zymogenic forms) and the TF-independent FVIIa variant (both active and "in silico" zymogenized forms), the authors concluded that the 170-loop was primarily responsible for the plasticity and/or rigidity and the N-terminal insertion modulated it. Also, the allosteric E/E* and Z/Z* equilibria were shown to be influenced by the 170-loop, but not the N-terminus insertion.
The results will certainly advance the thinking of the scientists in the field and I would recommend this for publication. As a minor comment, I would encourage the authors to present the heat maps of the DCCMs and examine them closely. It would be appropriate to have a comment on whether the “curiously” similar DCCMs were due to the use of replica exchanges at every ps that constantly stirred up MD trajectories.
Reviewer 2 Report
The work is very interesting and well-performed and analyzed.
Authors indicate in the Materials and Methods section that they are applying a cut-off of 12 Å. Although this value is enough for the Lennard-Jones interactions, it is not for the Electrostatics. I assume that they are using a PME based approach, but it is not indicated in the manuscript. In addition, being gromacs, the cutoff-scheme is expected to be "Verlet", isn't it?
Page numbering in the references is mixed: some lack the pages (ref 24, ...), the ending page (8, ...) or is relative (4, ...)
Reviewer 3 Report
The authors present an interesting research subject. They apply a T-REMD Molecular Dynamics (MD) approach to study a trypsin-like serine protease (SP) domain of the coagulation factor VIIa (FVIIa), particularly the influence of its inactivation on the structural and dynamical properties. They model the enzymatic function of FVIIa and variants taken from the PDB. The authors compare the MD results of three PDB structures and their truncated version built to mimic inactive zymogen precursors, to characterize the inactivation process.
They observe:
- a conformational change of the truncated trypsin structure that eventually adopts a trypsinogen-like conformation
- a collapse of the activation pocket in the truncated structure
- an increased flexibility in the activation loops in the truncated structure. Compactness on the enzymes is also reduced
- enhanced flexibility similarities between the mutated version of FVIIa and trypsin. Both structures are among the most compact ones
- little influence of the inactivation regarding the accessibility of the active site to a substrate
The authors conclude about the feasibility of their “in silico” zymogenation protocol.
However, the paper is uneasy to read for people unfamiliar with the biological systems studied in the manuscript. It also lacks too many details and precisions regarding the interpretation of the results, which make the paper not always convincing.
Here follow my comments.
1) p. 3, sentence above Table 1. I would rephrase the sentence to make the paper clearer (e.g., at page 6, when the authors talk about three systems) “Three SP domains and their N-terminal truncated versions were studied, i.e., FVIIa, FVIIaVYT, and Trypsin (Table 1).”
The authors could explain why they chose those systems. Are they the only ones available in the PDB? Apparently not if I believe the last sentence of page 6.
2) p. 3, Table 1 – Is it really necessary to mention, for each system in Table 1, that T-REMD is the simulation protocol? It is the same for all systems. It could thus be mentioned in the text, once for all.
3) p. 4, Section 3.1 - “Here we adopt a replica-exchange …. not readily defined”. The sentence might be better placed in Section 2.1.
Additionally, could the author comment on the difficulty to determine a collective variable?
4) p. 5, Fig. 1 – if I understand well, the authors plot the RMSD profile calculated with MD frames of trypsin-desIVG compared to the averaged MD structure of Trypsin (black lines), and the RMSD profile calculated with MD frames of trypsin-desIVG compared to the averaged MD structure of Trypsinogen (red lines). From the visual inspection of Fig. 1, they infer that the lower values of the red profile means that trypsin-desIVG is structurally closer to trypsinogen. As far as I see, the difference between the two profiles is around 0.2 Angstroms. Is such a low value really significant considering the large fluctuations of the profiles?
It is linked to the sentence in page 4 “… more closely resembling the simulated average structure of trypsinogen …” . Could the authors more precisely explain what structural elements are structurally closer to trypsinogen? Is it related to the arrows displayed in Fig. 1a?
5) p. 5 and Fig.2 – “…practically identical extended spatial region of high flexibility…” What the reader should look at? What does “practically identical” mean? Loops AL1 to AL3 are mentioned, but the residue range is not reported (see also top of page 2). Please provide details.
6) p. 5, last sentence - If crystal structures are obtained at very low temperatures and crystal packing cannot be modelled, what is the significance of the comparisons between simulated and experimental B factors?
7) p.6, Fig. 2a - One distinguishes black and red profiles. What do they stand for? Simulated and experimental? Also, I assume the RMSF profiles are reported for the Calpha atoms only.
8) p.6 – Section 3.2 - “… reveal that the activation pocket collapse …”. What should the reader look at in Fig. 2? I do not see anything convincing about the collapse of the activation pocket.
A clear figure emphasizing the location of the activation pocket, the activation loops, the 170-loop (already shown in Fig. 1), the 215-217 segment mentioned further in the text, … might be useful (e.g., in the Supplementary Information section)
9) p. 6 - Section 3.2 – “Comparisom with the trypsinogen-trypsin …. This process appears conserved …. [66]”. I would rephrase that part of the text, if I understand well, using something like “Such an increase is observed in all zymogen-enzyme pairs. It is thus expected to be shared by most trypsin-like zymogens …” and explain why the authors expect such a tranferability in the flexibility pattern.
10) p. 7 – “The characteristics of ….. are striking”. The authors must help the reader by being more specific/precise about what to look at in Figure 2 (Figure 2 is built from 15 sub-figures).
11) p. 7 – The oxyanion hiole mentioned in the text could be illustrated in a general picture (see comment #8).
12) p. 7, 3rd paragraph – “Thus the zymogen … cooperative process …”. Should the reader understand that the activation transition is cooperative due to energetic origins? If so, why/how the electrostatic and hydrophobic interactions interplay to generate a cooperative effect? Could the authors describe more precisely the cooperative process and its energetic and entropic contributions, and/or make a clear link with results presented elsewhere in the manuscript?
13) p. 8, Fig. 3 - I find the E-E*/Z-Z*analysis interesting.
- The authors should precise why they consider residues 195 and 57 as references to calculate distances? Is it because their relatively low flexibility? Can the reader refer to Fig. 1?
- Is it possible to analyze the MD trajectories to evaluate the energy barrier corresponding to the various conformation changes? If not, why? Occurrence frequencies of these changes would also be interesting. It would help the reader to quantify and understand the negligeable influence of the inactivation process on the 215-217 loop conformation.
14) p. 9, 2nd paragraph – How do the authors define “compactness” of an enzyme? They mention that trypsinogen is the least compact structure. Is it related to the gyration radius or is it related to the space probed by the PC1-PC2 data points? In the last case, the authors must explain why/how PC1-PC2 plots allow to quantify the compactness (or is it the rigidity?) of a structure. I must admit that the concepts flexibility, plasticity/rigidity, and compactness, are hard to differentiate from the information reported by the authors, and I am not convinced that Fig. 4 brings more information than Fig. 3 does. The PC1-PC2 plots result from the conformations sampled during the MD Simulations, like the distance-distance plots of Fig.3 do.
In addition, the authors order the systems according to their compactness but the order given is not convincing if one considers Fig. 4 only, e.g., why should Trypsinogen (said to be the least compact) be less compact than FVIIa-desIVG?
In other words, the words “unambiguous trends” ask for more details and precisions.
15) p. 11 – The authors mention the importance of solvation, and suggest to use specialized analysis methods. Why a direct analysis of the water distribution, e.g., in the proximity of I16 and D194, or the 215-217 segment, is not reported here?
16) Regarding the preparation of the systems, did the authors kept the crystal water molecules? If so, are these molecules stable during the MD simulations? If not, are the crystal sites occupied by water molecules?
Round 2
Reviewer 3 Report
The new manuscript version hes improved and the authors replied to the concerns raised in the first review. I have only two very short and minor comments,
(a) regarding the sentence added in section 3.4: "..."compactness is consistent conformational rigidity ..." Should it be rather read
" ..."compactness is consistent with conformational rigidity ..." ?
(b) I do not find the reply to comment #6 in the new version of the manuscript (is it really located at page 6 as announced by the authors?)